# Evaluation of the Effects of Instant Cascara Beverage on the Brain-Gut Axis of Healthy Male and Female Rats

**DOI:** 10.3390/nu16010065

**Published:** 2023-12-25

**Authors:** Paula Gallego-Barceló, Ana Bagues, David Benítez-Álvarez, Yolanda López-Tofiño, Carlos Gálvez-Robleño, Laura López-Gómez, María Dolores del Castillo, Raquel Abalo

**Affiliations:** 1Department of Basic Health Sciences, Faculty of Health Sciences, University Rey Juan Carlos (URJC), 28922 Alcorcón, Spain; paula.gallego@urjc.es (P.G.-B.); david.beniteza@urjc.es (D.B.-Á.); yolanda.lopez@urjc.es (Y.L.-T.); carlos.galvezr@urjc.es (C.G.-R.); laura.lopez.gomez@urjc.es (L.L.-G.); 2Associated R+D+i Unit to the Institute of Medicinal Chemistry (IQM), Scientific Research Superior Council (CSIC), Calle Juan de la Cierva 3, 28006 Madrid, Spain; 3High Performance Research Group in Experimental Pharmacology (PHARMAKOM-URJC), University Rey Juan Carlos (URJC), 28922 Alcorcón, Spain; 4High Performance Research Group in Physiopathology and Pharmacology of the Digestive System (NeuGut-URJC), University Rey Juan Carlos (URJC), 28922 Alcorcón, Spain; 5Food Bioscience Group, Department of Bioactivity and Food Analysis, Instituto de Investigación en Ciencias de la Alimentacion (CIAL) (CSIC-UAM), Calle Nicolás Cabrera, 9, 28049 Madrid, Spain; mdolores.delcastillo@csic.es; 6Working Group of Basic Sciences on Pain and Analgesia of the Spanish Pain Society (Grupo de Trabajo de Ciencias Básicas en Dolor y Analgesia de la Sociedad Española del Dolor), 28046 Madrid, Spain; 7Working Group of Basic Sciences on Cannabinoids of the Spanish Pain Society (Grupo de Trabajo de Cannabinoides de la Sociedad Española del Dolor), 28046 Madrid, Spain

**Keywords:** behavior, brain-gut axis, dried coffee cherry pulp, gastrointestinal motility, instant cascara, radiographic methods, rat, sex

## Abstract

Instant cascara (IC) is a sustainable beverage obtained from dried coffee cherry pulp, rich in nutrients and bioactive compounds. The present research aimed to determine the effects of IC on general health and brain-gut axis parameters of healthy female and male rats. Wistar rats were exposed to IC (10 mg/mL) in their drinking water for 3 weeks. Body weight and solid and liquid intakes were monitored as indicators of food safety. Gastrointestinal transit was radiographically evaluated one day (acute) and 3 weeks (chronic) after the start of IC exposure. Locomotor activity, anxiety, and anhedonia of the animals after 3 weeks of treatment was also studied. Overall, compared to water-exposed animals, IC significantly increased food intake in males (*p* < 0.0001) and liquid intake in females (*p* < 0.05) without changes in body weight in either case. IC did not significantly modify gastrointestinal motility parameters after its acute or repeated intake and did not cause any significant behavioral alterations in males or females (*p* > 0.05). In conclusion, repeated intake of IC at the studied concentration did not negatively affect brain-gut axis functions of healthy male and female rats. Anxiety behavior, diarrhea, constipation, abnormal weight modifications, or other typical effects of toxicity were not observed in animals treated with the new powdered beverage, suggesting its food safety under the studied conditions.

## 1. Introduction

During coffee cherry processing, approximately 90% is discarded, causing negative environmental consequences [1]. Coffee cherry pulp is amongst the main and first by-products generated in the coffee cherry processing. During the last years, an important effort has been made to produce different products from it because of its rich nutrient content and health-promoting properties [2,3]. In 2022, it was approved for its commercialization in the European Union [4]. Also, in the United States and in countries of Central and South America, they produce and sell beverages and bakery products using this coffee by-product [5]. In Europe, the commercialization of drinks made from coffee husks is authorized. To the best of our knowledge, dehydrated or powdered drinks made with that are not commercially available. Food powders are of great interest since they are the most common format of food materials. There are several reasons for this, such as low bulk weight, storage, transport and usage conveniences, diverse applications, relatively high stability, and the possibility of a high production rate. They are easy to use, pack, distribute, and handle [6].

Instant cascara (IC) powder is obtained by hot water extraction and freeze-drying from dried coffee cherry pulp. The nutritional composition of powdered IC has been previously reported [5]. According to the Annexes of Regulations (EC) n° 1924/2006 and n° 1169/2011 [7], it is declared as a “low-fat” drink, “without added sugars”, “high in fiber”, “source of potassium and magnesium” and “low in caffeine content” [8,9]. Moreover, it contains only low amounts of the toxic compound acrylamide, but it is a good source of melanoidins with potential health-promoting properties [7]. These types of products can contribute to satisfying the demands of nutritious, healthy, natural, and sustainable food and beverages made nowadays by consumers.

IC beverages can be prepared at various concentrations, in water, milk, etc. Because of its rich nutrient content and bioactive compounds, IC can also be employed as a food supplement. Food supplements are concentrated sources of nutrients or other substances with a nutritional or physiological effect, as it is the case for IC. They are marketed in dose forms (powder, pills, tablets, capsules, and liquid). Food supplements are not intended to modify physiological functions but to correct nutritional deficiencies, maintain an adequate intake of certain nutrients, or support specific physiological functions such as those of the brain-gut axis (BGA), which are very relevant for holistic human health [10,11,12,13]. To the best of our knowledge, studies reporting the effects of the regular intake of IC or other beverages derived from dried coffee cascara on the BGA have not been previously reported, although they are of great interest due to the increased popularity of these products after their approval as a novel food by the European regulatory authorities and their chemical composition.

The BGA communicates in a bidirectional way the central nervous system (CNS) and the gastrointestinal (GI) tract through the autonomic nervous system (ANS), the hypothalamic-pituitary-adrenal (HPA) axis, the enteric hormones, the signals derived from the gut immune system and the metabolites secreted from the gut microbiota [14,15,16]. Thus, this axis modulates different cerebral and GI functions, such as the response to stress, motility, permeability of the GI mucosa, colonic sensitivity, immune function, or the secretion of certain hormones [14,17,18]. The alterations in this axis can be the cause of GI functional alterations such as irritable bowel syndrome and contribute to symptoms of inflammatory bowel disease, which includes Crohn’s disease and ulcerative colitis [17,18,19,20]. Additionally, sex is one of the main factors that affect the cerebral and GI functions and determine the prevalence and severity of the diseases affecting the BGA [21,22,23,24]. Thus, the study of the BGA functions in both males and females, whether under healthy or pathologic conditions, is extremely important.

Considering the growing interest in the commercialization of IC beverages and the need for basic research to study the impact of coffee fruit cascara in the physiology of the organism, as well as the importance that sex dimorphism has in the normal functioning of the BGA and may have in the appearance and severity of different pathologies in this system, we sought to investigate the possible in vivo effects of the beverage IC on the BGA of healthy male and female rats.

## 2. Materials and Methods

### 2.1. Ethics Statement

The experimental protocol was approved by the Ethics Committee of Rey Juan Carlos University (URJC) and Comunidad de Madrid (PROEX-059/2018) and complied with the European Community Council Directive of 22 September 2010 (2010/63/EU) and Spanish (Law 32/2007, RD 53/2013, and order ECC/566/2015) regulations, for the protection of animals intended for scientific studies. The protocol design aimed to minimize animal suffering and discomfort as well as to reduce the number of animals used.

### 2.2. Instant Cascara Beverage

Coffee cascara from Arabica species and Tabi variety from Colombia was provided by SUPRACAFÉ S.A. (Móstoles, Madrid, Spain). Coffee cascara was obtained using the wet method, which consists of a phase of pulping and drying in the sun over a period of one week, in which the humidity is reduced to 10%. Finally, a sanitization process is carried out using irradiation (IONISOS Ibérica).

Powdered extract from coffee fruit cascara was obtained by an aqueous extraction of 50 g/L at 100 °C for 10 min (extraction yield was 20%) [25]. The sample was filtered (250 μm) and freeze-dried to obtain the powder called IC. The nutritional composition of IC powdered extract reveals significant macronutrient content, including 47.48% carbohydrates, 18.32% dietary fiber, 0.58% lipids, and 6.25% proteins. Among the prominent micronutrients, IC showcases elevated levels of potassium (6701 mg/100 g), magnesium (121.56 mg/100 g), sodium (354.19 mg/100 g), and calcium (109.88 mg/100 g). Furthermore, the presence of ascorbic acid is noted at a concentration of 438.95 mg/100 g. The bioactive compounds within IC encompass caffeine (1.39%), chlorogenic acids (10.7–12.6%), and melanoidins (15%) [5].

### 2.3. Animals and Experimental Groups

Male (250–350 g) and female (200–250 g) young adult (2–3 months) Wistar rats obtained from the URJC Veterinary Unit were used in the study. Animals were kept in standard transparent cages (60 cm × 40 cm × 20 cm), separated by sex, and in groups of 3 rats per cage. The animals had access to chow pellets, a SAFE D40 diet (www.safe-diets.com, accessed on 27 May 2021), and sterile tap water “ad libitum”. Temperature (20 °C) and humidity (60%) were controlled and constant. In addition, 12/12-h light/dark cycles were used, with the lights on during the day (from 8:00 to 20:00).

Rats were randomly assigned to four experimental groups based on sex and the beverage, i.e., half of the male and female animals received normal water in their drinking bottles, whilst the rest were exposed to IC beverage (IC powder was added to water at a concentration of 10 mg/mL in the bottle) as the only available beverage. The concentration was chosen based on a previous study [9].

### 2.4. Experimental Protocol

Two different cohorts of animals were used in this study. The first cohort of animals (*n* = 12 goup) were exposed to water or IC beverage for three consecutive weeks. Weight, solid, and liquid intake were recorded throughout the study. During the third week, anhedonia and exploratory behavior were assessed using the splash and hole board tests, respectively. GI motility was evaluated by radiographic methods one day after the behavioral assessments. At the end of the third week, the animals were sacrificed, and the organs were obtained for macroscopic analyses. In females, a vaginal cytology was performed to determine the phase of the estrous cycle before the different studies and the sacrifice (Figure 1).

In the second cohort (*n* = 9/group), a radiographic study was performed after the first day of IC beverage exposure (acute effects), and thereafter, the animals were returned to their cages, and exposure to IC or water was maintained for 3 weeks. During the last week of exposure, anxiety was assessed with the plus maze test (Figure 1). Vaginal cytology was also performed in the female rats of this cohort.

All analyses were performed by experienced researchers who were blinded to the treatments.

### 2.5. Body Weight, Solid and Liquid Intake

Body weight and solid and liquid intake were measured manually three days a week (Monday, Tuesday, and Friday) during the three weeks of IC or water exposure and just before sacrifice at the end of the study. IC beverages were not reused, and leftovers were discarded. Food and liquid intakes are represented as the mean intake per rat and day for each experimental group.

### 2.6. Vaginal Cytology Smear in Female Rats

Vaginal smears were performed in females at different time points: before carrying out GI motility experiments, after the behavioral ones, and before sacrifice. A cotton-tipped applicator was rotated three times at 2 cm from the vaginal orifice. The vaginal smears were placed on slides, stained with hematoxylin and eosin, and observed microscopically to determine the phase of the estrous cycle [26]. The estrous cycle in the rat is divided into proestrus, estrus, metestrus, and diestrus [26]. Proestrus is characterized by a predominance of nucleated epithelial cells (corresponds to the follicular state in humans and is related to an increase in the concentration of circulating estradiol), estrus by non-nucleated cornified epithelial cells (in human female cycle corresponds to the decline in estradiol), metestrus displays an equal proportion of nucleated or non-nucleated epithelial cells and leukocytes, and diestrus a predominance of leukocytes (these last two phases display high levels of progesterone and correlate, respectively, with the early and late secretory phases of the human female cycle) [26].

### 2.7. Behavioral Tests

#### 2.7.1. Splash Test

The splash test is a behavioral assay aimed at detecting a possible state of anhedonia in rodents. The splash test was performed, as described by Jacenik et al. 2021 [27]: a 10% sucrose solution was sprayed on the back of the animal, which responded with a grooming behavior. Grooming includes nose and face stroking (e.g., stroking along the snout), head grooming (e.g., semicircular movements of the paws over the top of the head and behind the ears), and body grooming (licking the body fur). After spraying the sucrose solution, the latency and duration of the grooming behaviors were recorded for 5 min.

#### 2.7.2. Hole Board Test

The hole board test was performed after the splash test to determine exploratory behavior, neophilia, and neophobia. Neophilia is defined as the attraction that animals display towards a novel object or place, whilst neophobia is the aversion that animals show towards an unknown object or place. The hole board consists of a box (60 × 60 cm) with a slightly raised floor divided into 36 squares (10 cm × 10 cm) and has 4 holes (4 cm in diameter) equidistant from each other and from the walls (40 cm high). The interior of the hole board was cleaned with 0.1% acetic acid and allowed to dry between trials. Each rat was removed from its home cage and placed in the same corner of the hole box, and its behavior was recorded for 5 min using a video camera (Panasonic HC-V160). The researcher was not present while the recording took place. Subsequently, the videos were analyzed, and the following parameters were evaluated: the number of external and internal squares crossed by the rat (spontaneous locomotion); the amount of time that the rat placed its head in one of the holes, at a minimum depth such that the ears were at the same level as the floor of the box (head-dip); the amount of time the animal remained motionless and lifted its front limbs from the ground, standing vertically on its hind limbs; the amount of time the animal spent grooming; and the number of feces expelled during the test. Each rat was subjected to this test once [27].

#### 2.7.3. Elevated Plus Maze

To determine possible anxious behavior, animals were placed in the elevated plus maze. It is a plus-shaped platform with two opposite open arms (50.8 cm × 10.2 cm) and two closed arms (50.8 cm × 10.2 cm × 40.6 cm), which are elevated 72.4 cm above the floor. Each animal was placed in the center of the maze and allowed to explore it for 5 min. As in the previous case (hole board test), the researcher was not in the room while testing, and each experiment was video recorded. The recordings were afterward visualized, and the following parameters were analyzed: the total time spent in open arms and in closed arms; and the number of times that the animals entered the open arms [28].

### 2.8. Gastrointestinal Motility (Radiographic Study)

General GI motility was studied using radiographic methods one day after initiation of IC beverage exposure and during the third week of the study (one day after performing the behavioral studies, Figure 1). Each rat was administered 3 mL of a barium sulfate suspension by gavage (Barigraph^®^AD, Juste SAQF; 2 g/mL). Radiographs of the GI tract were obtained at different time points after barium administration: immediately (0) and at 1, 2, 3, 4, 6, 8, and 24 h. Radiographs were performed using a CS2100 digital X-ray device (Carestream Dental, Madrid, Spain; 60 kV, 7 mA) and recorded on a digital plate (Venu1717V, iRay Technology, Madrid, Spain), sensitive to radiation, which allows automatic image processing in the computer. The exposure time was set to 20 ms, and the focus distance was manually set to 50 ± 1 cm. The rats did not receive anesthesia and were immobilized in a prone position inside transparent, handmade, and adjustable plastic tubes. Previously, the rats were habituated to stay inside the tube, and they were returned to their cage immediately after each X-ray shot (immobilization lasted for up to two minutes). Alterations in GI tract motility were semiquantitatively determined from the X-ray images by assigning a compounded value to each GI region (stomach, small intestine, caecum, and colorectum) considering the following parameters: percentage of the region filled with contrast (0–4); intensity of contrast (0–4); homogeneity of contrast (0–2); and sharpness of the gut region profile (0–2). Each of these parameters was scored, and a sum (0–12 points) was calculated [29]. The size, as well as density of the stomach, cecum, and fecal pellets, were analyzed using an image analysis system (ImageJ 1.38 for Windows, National Institute of Health, USA, free software: https://imagej.nih.gov/ij/ (accessed on 8 January 2021)) [27].

Once the first radiograph was taken after barium administration (0), the rats were placed in new cages with fresh bedding. Feces expelled by the rats belonging to each cage were collected during the remaining time points of the radiographic session (1, 2, 3, 4, 6, 8, and 24 h). Feces were analyzed to determine the degree of humidity and the total GI transit (measured as the moment barium appeared in feces), and the following parameters were measured: wet weight, dry weight (after drying the feces in an oven at 70 °C for 24–48 h), weight difference (wet-dry), total number of feces, and index of marked feces (the feces were X-rayed and scored according to their level of staining due to barium: low-marked stools visualized on the radiograph at the different time points were multiplied by a value of 1, and the strongly marked stools by a value of 2; the sum of these two factors was divided by the total number of stools, produced by each rat, at each time point). These parameters were represented as the average per rat and per number of hours elapsed since the previous time point [30].

### 2.9. Macroscopic Analysis

At the end of the third week, animals were sacrificed under anesthesia with sodium pentobarbital (2 mL/kg). The stomach, small intestine, cecum, and colon were removed *en bloc* and extended over a sheet of graph paper, and a photograph was taken, which was used for the macroscopic study. The images obtained were analyzed with ImageJ to determine the area of the cecum and the stomach and the length of the small intestine and colon for each rat [27].

Once the photograph was taken, each organ was weighed separately on a precision scale. The small intestine and colon were weighed after the removal of the contents. The small intestine was milked, and its content also weighed [27]. In addition, epididymal/periovarian and retroperitoneal fat pads were extracted and weighed.

### 2.10. Data Analysis

Statistical analyses were performed with the GraphPad Prism 8 program. All data passed the Shapiro-Wilk’s normality test. The results of vaginal cytology were expressed as % of females in each phase of the estrous cycle and analyzed using the chi-square test. The other results were expressed as the mean of the values obtained ± SEM (standard error of the mean) and were analyzed using a one-way or two-way ANOVA, followed by Bonferroni’s post-hoc test for multiple comparisons. Differences were considered statistically significant when *p* < 0.05. In all graphs, the # and $ symbols are utilized to denote disparities arising from the sex of the animals (males vs. females; # without and $ with IC), while the * and + symbols are employed to signify variances attributed to the beverages administered (control vs. IC beverage; * in males and + in females).

## 3. Results

### 3.1. Body Weight, Solid and Liquid Intake

Throughout the three weeks of the study, the weight of the male and female control rats was expected for their age (2–3 months old), and the intake of IC did not significantly (*p* > 0.05) alter these results with respect to the control animals.

With regards to solid intake, overall, females ate less than males (*p* < 0.0001). The male-IC group presented a statistically significant higher average solid intake compared to their control group (*p* < 0.0001). In females, however, IC exposure did not significantly alter the average food intake (Figure 2B).

Regarding liquid intake, females also had a statistically significant lower average intake compared to males (*p* < 0.001). Both in males and females exposed to IC, the liquid intake increased when compared to their respective sex-matched controls, although the difference was only statistically significant in females (*p* < 0.05) (Figure 2C).

### 3.2. Behavioral Tests

#### 3.2.1. Splash Test

In the splash test, two parameters were evaluated: the time that the animals took to start grooming (latency) and the time they spent grooming (duration) (Table 1). Both parameters were similar in male and female animals, and the exposure to IC did not significantly modify these results (*p* > 0.05).

#### 3.2.2. Hole Board Test

Table 2 shows the behavioral results of the hole board test. The locomotor activity (total number of squares traveled) and exploratory behavior (time spent at the holes) of control females and males did not show statistically significant differences (*p* > 0.05). IC intake tended to increase the time females stood on hind limbs, although this difference was only significant when compared to males-IC (*p* < 0.05).

#### 3.2.3. Elevated Plus Maze

As shown in Table 3, the time that control animals spent in the closed and open arms of the elevated plus maze was similar in males and females, and IC beverage did not significantly modify these parameters (*p* > 0.05). The number of entries into the open arms was also similar in all groups, without statistically significant differences amongst them (*p* > 0.05).

### 3.3. Gastrointestinal Motility (Radiographic Study)

Gastrointestinal motility was evaluated using radiographic methods at two time points (in different cohorts of animals, as mentioned in the Methods section): after just 1 day (cohort 2) and after 3 weeks (cohort 1) of IC beverage exposure. Different analyses were performed on the X-rays and the feces obtained during the radiographic sessions.

#### 3.3.1. Semiquantitative Analysis of Gastrointestinal Motility

In the 24-h study, gastrointestinal motility was similar in all regions among the different experimental groups, and only minor significant differences (*p* < 0.05) were observed at the end of the radiographic session, showing a slight delay in gastric emptying in females vs. males among the IC-treated groups (at T8) and a slight delay in cecum emptying in females vs. males among the control groups (at T24) (Appendix A).

Regarding the chronic study, statistically and clinically relevant differences were found among groups, also associated with sex, but not IC (Figure 3). Gastric emptying (Figure 3A) was progressive in both sexes after the administration of barium until the end of the study and faster in females than in males. Transit through the small intestine was similar across groups except for a slightly faster emptying at T3 in females (Figure 3B). Accordingly, the cecum of females started to fill up faster and reached statistically significant higher values than those of control males from T4 until T6, with similar values during the rest of the study among groups (Figure 3C). Transit across the colorectum occurred from T3 until the end of the study in a similar manner across groups, but at T24, more barium was observed in females than males (Figure 3D). Likewise, the number of feces in the colorectum was similar throughout the study in all groups, and only an increase in females when compared to males was observed at T24 (Figure 3E). IC beverage did not significantly change any parameter with respect to the controls in any of the studied organs.

#### 3.3.2. Morphometric and Densitometric Analysis of Gastrointestinal Organs

The morphometric and densitometric analysis of the stomach, cecum, and fecal boluses showed similar changes over time to those found in the semi-quantitative study. However, these analyses provide additional information on the maximum size and density of the GI organs and fecal boluses, as highlighted in the following descriptions.

In the acute study, no statistically significant differences were found among groups for the maximum size or density of the stomach (Appendix A). The maximum size of the cecum (Appendix A) was significantly reduced in females compared to males, and IC did not significantly alter these results, whilst the maximum barium density in this organ was not altered by either sex or IC exposure (Appendix A). The maximum size of fecal pellets within the colorectum was overall smaller in females than males, and the difference was statistically significant between the IC-treated animals (females vs. males, Appendix A). The maximum barium density in the fecal pellets was similar in all groups, without statistically significant differences among them (Appendix A).

The maximum size of the stomach after 3 weeks of treatment was significantly smaller in control females than males (Figure 4A). The maximum areas of the stomachs of the groups exposed to the IC beverage were larger compared to their corresponding controls, but the differences were only statistically significant for males. Furthermore, the maximum density of the stomach (Figure 4B) was close to 90% in all groups and showed statistically significant differences due to sex, with females (control or exposed to IC) displaying higher densities than males, irrespective of beverage exposure.

Compared to control males, the maximum area of the cecum was significantly smaller in control females (Figure 4C). The IC beverage did not significantly modify the maximum cecal areas of males and females. On the other hand, the maximum density (Figure 4D) was also significantly greater in females compared to males. The IC drink did not modify the maximum density of the cecum in either sex.

The feces of females were also significantly smaller (Figure 4E) and had a significantly greater density (Figure 4F) than those of males. Exposure to IC did not cause any statistically significant difference in these parameters in either males or females.

#### 3.3.3. Analyses of the Feces Collected during the X-ray Session

The feces collected during the two X-ray sessions, after acute and chronic exposure to IC beverage, produced some interesting results, shown here for number (Figure 5A,B), wetness (Figure 5C,D), and index of stained fecal pellets (Figure 6A,B).

In both the acute and chronic studies, control females presented statistically significant lower values of fecal number and wetness at T1 than control males; during the rest of the radiographic session, the values continued to be lower than those for control males but without statistically significant differences, because the values in the male groups dropped considerably from T1 to T2. The IC groups did not show any statistically significant differences compared with their respective controls (Figure 5A–D). However, during the acute study, IC displayed a trend of increased defecation and fecal wetness from T1 to T6 in males (Figure 5A,C) that was not observed in the chronic study (Figure 5B,D). Similar trends were found for wet and dry fecal pellet weight (Appendix A).

In the acute study, the index of stained fecal pellets (Figure 6A) showed that labeled stools started appearing in the cages at T4 in all groups except females exposed to IC beverage and increased gradually throughout the study, peaking at T8–T24, depending on the group, without any significant differences among them; no group showed an average staining index higher than 1 (Figure 6A). In the chronic study, the appearance of barium staining was from T4 in all groups, and average values peaked at T8–T24, reaching 1.3–1.5 points for all of them (Figure 6B). In the control groups, females presented a similar index to control males throughout the study. Although the IC beverage did not significantly modify the curves observed in this parameter compared with those obtained for their controls, they were slightly faster for the males, whereas in females, IC beverage slowed down these curves, making these differences statistically significant at T6–T8 when compared to the group of males exposed to IC.

### 3.4. Macroscopic Analysis

The macroscopic characteristics at the sacrifice of each gastrointestinal organ and the fat of the rats from cohort 1 are shown in Appendix A. The differences observed in the parameters evaluated were mainly due to sex (even after normalization to body weight). Only the cecum area and the length of the colon measured before removing the feces were significantly different (bigger) in males treated with IC compared with control males.

### 3.5. Vaginal Cytology Smear

The distribution of female animals according to the estrous cycle in which they were found at the 3 different time points of the study can be observed in Table 2. No statistically significant difference was observed between control and IC-exposed females in the results obtained after the vaginal cytology performed after the behavioral assays or before sacrifice. However, in the gastrointestinal acute study (cohort 2), the distribution of females according to the estrous cycle showed significant differences (*p* < 0.05) between the control and IC-treated groups.

## 4. Discussion

Commercially available beverages such as tea, mate, and coffee contain similar compounds (fiber and caffeine) to those in IC beverages [31,32,33,34]. However, IC is a novel powdered drink that differs from conventional coffee beverages and other coffee husk drinks. The freeze-drying or dehydration process used to create IC helps preserve and concentrate the natural nutrients and bioactive compounds found in the coffee cherry husk. As a result, IC presents a higher concentration of vitamins, minerals, antioxidants, and beneficial compounds compared to other commercially available beverages made from coffee husk [9]. With its unique processing method that preserves nutritional value, IC offers a distinctively nourishing and wholesome choice among coffee husk beverages [33,35,36]. Due to its properties, it might be interesting to propose its use as a food supplement or new product with an increased concentration, but before doing this, it is necessary to clearly delineate their potential beneficial or deleterious effects on the organism. Therefore, it is essential to conduct animal studies to ascertain the safety of these beverages in a controlled and ethically sound environment before proceeding with clinical investigations.

Thus, the aim of the present study was to determine the possible effects of regular intake of IC at a dose of 10 mg/mL, freely soluble in water, on the normal physiology of the BGA in male and female rats through different behavioral tests and GI transit techniques. In our study, most changes observed among the experimental groups were due to sex. Regular IC intake only induced significant increases in the average food intake in males (which were associated with slight but significant increases in stomach and cecum size and colon length) and liquid intake in females (without morphometric alterations). Importantly, however, anxiety behavior, diarrhea, constipation, abnormal weight modifications, or other typical effects of toxicity were not observed.

Sex-related differences in the BGA have been reported in a wide variety of articles [24,35,36,37,38,39]. These differences need to be taken into account to establish an adequate treatment of the symptoms of functional GI disorders, anxiety, depression, or neurological diseases [18]. However, sex does not only influence different aspects related to pathology but also different parameters in healthy individuals, as observed in this study and previous ones [24].

Compared to males, females had lower body weight gain and lower food and liquid intake, with no differences in normalized weight for epididymal/periovarian and retroperitoneal fat. In addition, female rats showed a reduction in the size of the GI organs compared to male rats (observed in the chronic radiographic study and confirmed in the macroscopic study at the end of the 3-week study). All these differences are expected, considering the sex-dysmorphic biology of these rodent species, as previously described [40,41].

In the healthy animals used in this study, IC beverage increased solid (males) and liquid (females) intakes but did not modify body weight gain or epididymal/periovarian or retroperitoneal fat deposition relative to their corresponding controls. The higher food intake in males exposed to IC was associated with a slight but significant increase in stomach and cecum size and colon length, suggesting that the higher load of food intake may cause morphometric adaptations of the gastrointestinal organs, which are well known for diabetic animals displaying hyperphagy [42] but have not been well defined in otherwise healthy individuals. Whatever the case may be, the maintenance of body weight in males, despite higher food intake, could be due to higher energy expenditure (greater activity) and/or greater fecal excretion, as suggested by the trends observed in studies of behavior (hole board test) and of feces (during the radiographic study). Although more specific studies are required to clearly determine the contribution of each factor, our results agree with other reports in which the addition of coffee, coffee grounds, or melanoidins in the diet of male Wistar rats did not significantly alter body weight [43,44,45,46]. On the other hand, the higher intake of liquid in IC-exposed than in water-exposed females did not modify body weight gain either and did not seem to significantly alter any other parameter studied here. The difference found in IC intake may be associated with sex-related differences in the sensory system in rodents [39]. Indeed, sexual dimorphism significantly shapes taste-guided behaviors observed in both rodents and humans, leading to discernible variations in preferences and responses between males and females [47,48]. The manipulation of sex hormones, particularly estrogen, exerts a profound influence on the comprehensive restructuring of these behaviors, spanning from central taste processing to peripheral nerve responses [48]. Nevertheless, the higher intake of IC in females compared with males did not exert any significant impact on any of the parameters tested here. Whether or not other functions could be affected, maybe after more prolonged exposure, needs to be analyzed, but our present results suggest this beverage does not critically affect female (or male) health.

After the first 24 h of IC exposure, the radiographic study showed that motility in the stomach, small intestine, cecum, and colorectal region was similar in all animals, regardless of sex and drink, whilst in the chronic study, females presented a slightly accelerated motility of the upper GI tract, compared to males, which is similar to the results obtained in previous studies [24]. In both studies, the control males presented similar GI transit patterns and maximum sizes of stomach, cecum, and fecal boluses that were also similar to those observed in previous studies [29,45,46]. The slightly accelerated transit in females in the chronic study might be related to the observed morphometric differences in the GI organs, which were significantly smaller than in males in this cohort at the time point evaluated (3 weeks after initiation of the study). In contrast, in the cohort used for the acute study, there were no morphometric differences in the stomach, although the differences between females and males in the areas of caecum and fecal boluses were similar to those encountered in the chronic study (but slightly lower). Although it cannot be discarded that these morphometric differences (especially in stomach size) between the two radiographic studies in female animals might be due to the three weeks elapsed, they seem more likely due to the fact that two different cohorts were used in these studies. Furthermore, in control females, significant differences were observed in the distribution of phases in the estrous cycle in the acute study compared to the chronic one (%: proestrus 17 and estrus 83 in cohort 1 vs. proestrus 33, estrus 17, mestestrus 42, and diestrus 8 in cohort 2, *p* < 0.05, Chi-square test), meaning that this factor might have contributed to the observed differences. Gastrointestinal motor function has been traditionally studied only in males or in mixed groups of female and male experimental animals [24]. Our results further support the need to evaluate gastrointestinal functions in both sexes separately and to take into account the estrous cycle phase in females.

As mentioned above, in both acute and chronic studies, the maximum size of the boluses was significantly lower in females than in males. Furthermore, despite the shorter length of the colon in females, the number of fecal boluses seen in it increased in females at the end of the study, suggesting a retention in this organ. These results are similar to those obtained in other studies performed in our laboratory [24]. Despite the fact that estrogens and androgens are important modulators of colonic smooth muscle contraction and motility [49,50], the differences in colonic transit occurred without a clear influence of the estrous cycle (the variability observed in these data was as low as in males) and only from T8 to T24, a period that included the circadian phase of the day in which rodents are more active (lights were off from 8 pm to 8 am), suggesting that the circadian rhythm (or the feeding and activity behavior related with it) affects the colonic transit of females and males in a different manner, although the mechanisms involved remain to be elucidated [24].

In general, the characteristics of the feces excreted during the radiographic study in control males presented similar results to those previously obtained in our laboratory in rats [24] and mice [30]. Compared to males, females presented lower values in the number of feces (Figure 5A,B), dry and wet weight (Appendix A), and wetness (Figure 5C,D) throughout both the acute and chronic studies, possibly related to the natural sexual dimorphism of these animals: females have less body weight, their intakes are smaller, and, therefore, they produce fewer feces [41]. However, the greatest (and statistically significant) difference between the sexes in fecal output and fecal moisture occurred at T1. At this time point, the higher values found in males in these parameters are possibly due to the initial stress associated with handling (due to the intragastric administration of barium) and the new conditions (new cages and sawdust) [51]. The present and previous results in rats [24] and mice [30] suggest that female rodents could be more resilient to this initial stress than males, at least during the non-active phase of their circadian rhythm [24].

Regarding the effect of IC in GI transit and the characteristics of the fecal boluses in the radiographic study, the only statistically significant difference was the stomach size, which was bigger in Males-IC than in Males-control. In the macroscopic study, only statistically significant increases in the areas of the cecum and colon with boluses in IC-males were observed. Despite being one of the most widely consumed beverages, there are relatively few in vivo preclinical studies on the effects of coffee and its derivatives on GI motility [52]. In humans, coffee has been shown to induce propulsive activity and colonic motility, especially in women [53,54]. Although caffeine produces direct motor effects on the GI wall [55,56,57], it is present only in a low proportion of the IC beverage [5]. Another component that can participate in the effects of coffee is dietary fiber (including melanoidins) due to its laxative effect and the production of short-chain fatty acids (SCFA), which are obtained from the fermentation of fiber by the intestinal microbiota and may stimulate colonic motility [52,55]. In previous studies, coffee by-products with a high proportion of fiber, like coffee grounds and melanoidins from coffee husks, increased GI transit in male Wistar rats [45,46]. Interestingly, in the present study, a similar but non-significant trend was found at first exposure of males to IC (acute study). However, this trend of increased defecation in males completely disappeared with chronic exposure to IC beverage, suggesting that tolerance may occur to its pro-motility effects, as also found for other coffee derivatives [46]. Although it cannot be ruled out that methodological differences may partly account for the differences, the scarce effects of the IC beverage on GI motility observed in vivo in the present study could be explained, at least partially, by the lower concentration of fiber in the IC powder (18.32% [5]) compared to that of coffee grounds (41.6%, [45]) and melanoidins (75%, [46]). Furthermore, our motility results align with those observed for other beverages similar to IC. For example, a study conducted on male rats subjected to mate beverage administration for a 2-week period did not reveal any alterations in intestinal propulsion [58]. On the other hand, in the case of tea, specifically green tea, the administration for a period of 30 days to male rats had significant effects on gastric emptying and small intestine propulsion, consequently enhancing GI motility [59]. This is attributed to the presence of tea flavonoids. In contrast, IC beverages lose the flavonoids during the drying phase of the coffee husk, and this may explain why these effects are not observed [9].

In terms of behavior, some differences between sexes were also observed, although, in general, they were not significant. We did not observe any anhedonic or anxiogenic behaviors in the splash and plus-maze tests, which are specific validated tests for the analysis of these behaviors [60,61]. Palumbo et al., 2020 reported sex-related differences in behavior in situations such as anxiety or depression, which are influenced by the regulation of the HPA axis, but they also found that control females tended to present an increased locomotor activity and a reduction in anxiety-related behaviors when compared to control males, whilst no sex-differences were observed in the anhedonic behavior [62]. Also, gonadal hormones (estrogen and androgens) can affect physical activity [63], and Ogawa et al., 2003 showed that the administration of estradiol benzoate increased the running wheel activity of male and female mice, possibly through activation of the estrogen receptor-α [64]. In our study, although the differences did not reach statistical significance, compared to control males, control females showed higher reactivity in the splash test and higher general exploratory activity in the hole board test, which is in accordance with previous studies performed in rats [65], but a lower defecation rate (which agrees with the results of the fecal pellet analysis during the first hour of the radiographic study). Importantly, the variability of the responses in both groups (control males and females) was similar, suggesting a low, if any, impact of the estrous cycle on our behavioral results. These results are like those previously obtained in our laboratory, also using rats of both sexes but fed with different purified diets [27].

The IC groups barely showed differences with respect to their controls in the behavioral studies, although, in the hole board test, they tended to have a greater horizontal exploration in males and vertical exploration in females. In male Wistar rats exposed to different diets supplemented with coffee, caffeine, or products of the Maillard reaction, no statistically significant differences were observed with the animals subjected to control diets in the motor and exploratory activity tests [44,66,67]. Although the use of more specific tests could detect more subtle effects of the IC beverage in animals, our results using the plus maze test and the splash test suggest that it is a product with little capability to induce anxiety or anhedonic behaviors, at least in healthy individuals, irrespective of their sex. Interestingly, despite mate beverage contains a higher quantity of caffeine compared to the IC beverage, chronic exposure to mate beverage in male mice did not result in significantly increased locomotor activity or anxiogenic effects when assessed through open field and elevated plus maze tests [68,69,70]. In contrast, the prolonged administration (for two weeks) of rooibos herbal tea (which lacks caffeine) in male rats resulted in an increase in their overall locomotion, as observed in the hole-board test [71]. Additionally, it enhanced their exploratory activity, measured by head tilting and rising behavior, while reducing anxiety-related behaviors, as evidenced by decreased thigmotaxis [71]. It is known that phenolic compounds in rooibos tea can influence brain amino acids, subsequently affecting behavior and neuroprotection [72]. The differences with our study may be attributed to disparities in the nutritional composition of the beverages.

In the present study, we have exposed the animals to IC for 3 weeks. Although many studies have used this time frame or even shorter [73,74], in other studies, the time of exposure is longer [45,46,75,76]. Thus, there is no homogeneity in the exposure time in the in vivo preclinical studies of the effects of beverages. Possibly, a more prolonged time of exposure to IC could have caused more intense effects.

Another possible limitation of the study is that animals were exposed to IC through drinking water and were kept grouped throughout the study. We did not administrate IC through gavage; thus, we cannot know the exact amount each rat drank. Although, on some occasions, gavage can be used, it has been demonstrated that the repeated use of this technique can have important effects on body weight gain and behavioral assays [77,78,79]. On the other hand, the animals were not isolated in metabolic cages (which would have provided more accurate solid and liquid intake data) to avoid alterations associated with isolation-induced stress. Indeed, isolated animals can get stressed, and therefore, this would also interfere with our study [80,81,82].

## 5. Conclusions

To conclude, this study has evaluated, for the first time, the effects of IC (10 mg/mL) on parameters related to the brain-gut axis in healthy male and female rats. Overall, most of the differences observed in this study were due to sex dimorphism, and only very few and small differences, with questionable clinical impact, were observed due to IC.

Thus, this study suggests that the “ad libitum” regular intake of IC beverages (10 mg/mL) does not modify health parameters related to the brain-gut axis of healthy animals.

## Figures and Tables

**Figure 1 nutrients-16-00065-f001:**
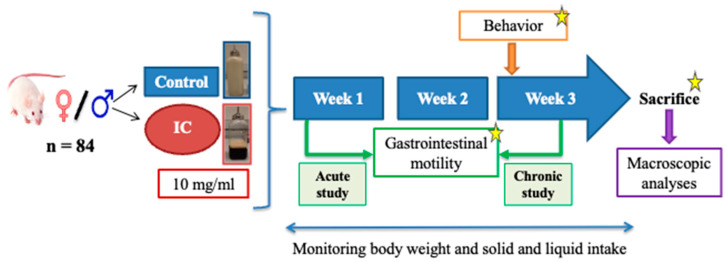
Experimental protocol. All rats were exposed to water (control group) or instant cascara (IC) beverage for 3 weeks, independently of the cohort of animals to which they were allocated. During the 3 weeks, body weight and food and drink intakes were regularly monitored. During the third week, different behavior assays were performed to analyze anhedonia (splash test, cohort 1), exploratory behavior (hole board test, cohort 1), and anxiety (plus maze test, cohort 2). Two radiographic motility studies were performed: after one day of IC beverage exposure (acute study, cohort 2) and after 3 weeks of IC beverage exposure (chronic study, cohort 1). At the end of the third week, animals were sacrificed, and a macroscopic analysis of the gastrointestinal organs was performed (cohort 1). The star represents the vaginal cytological smear evaluations performed to determine the phase of the estrous cycle in females in the indicated studies.

**Figure 2 nutrients-16-00065-f002:**
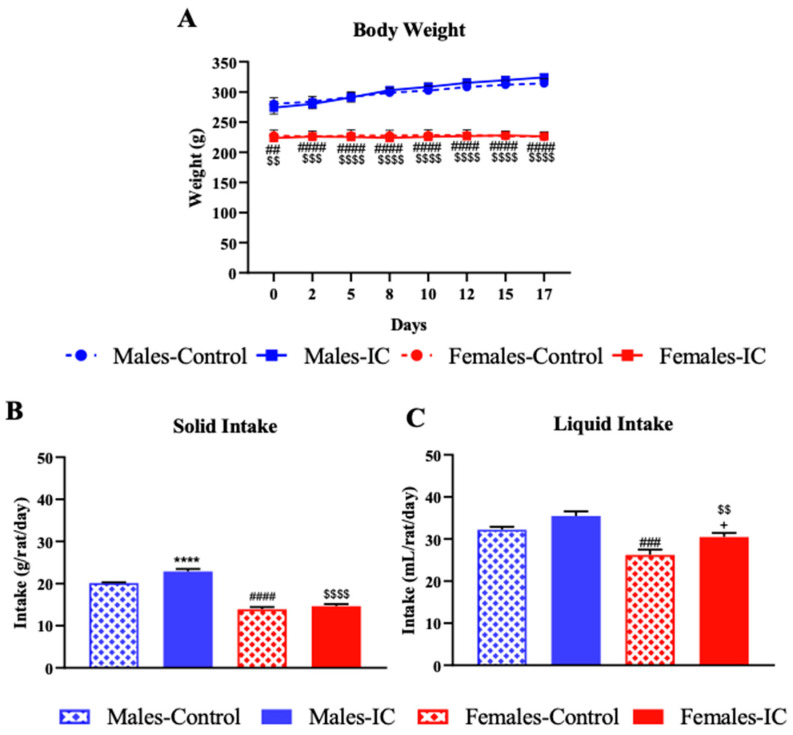
Effect of instant cascara (IC) beverage on the evolution of body weight and on the solid and liquid intake of rats of both sexes. The evolution of body weight (**A**) and the overall mean solid (**B**) and liquid (**C**) intake by each experimental group are shown. These parameters were recorded in the four experimental groups, distributed according to sex and the administered beverage (IC or water): Males-Control, Males-IC, Females-Control, and Females-IC. Data represent mean ± SEM (standard error of the mean). *N* = 12 animals per group (cohort 1). Sex-dependent statistically significant changes: ## *p* < 0.01, ### *p* < 0.001, #### *p* < 0.0001 (Females-Control vs. Males-Control); $$ *p* < 0.01, $$$ *p* < 0.001, $$$$ *p* < 0.0001 (Females-IC vs. Males-IC). Beverage-dependent statistically significant changes: **** *p* < 0.0001 (Males-IC vs. Males-Control); + *p* < 0.05 (Females-IC vs. Females-Control). (**A**): Two-way ANOVA followed by Bonferroni’s post-hoc test. (**B**,**C**): One-way ANOVA followed by Bonferroni’s post-hoc test.

**Figure 3 nutrients-16-00065-f003:**
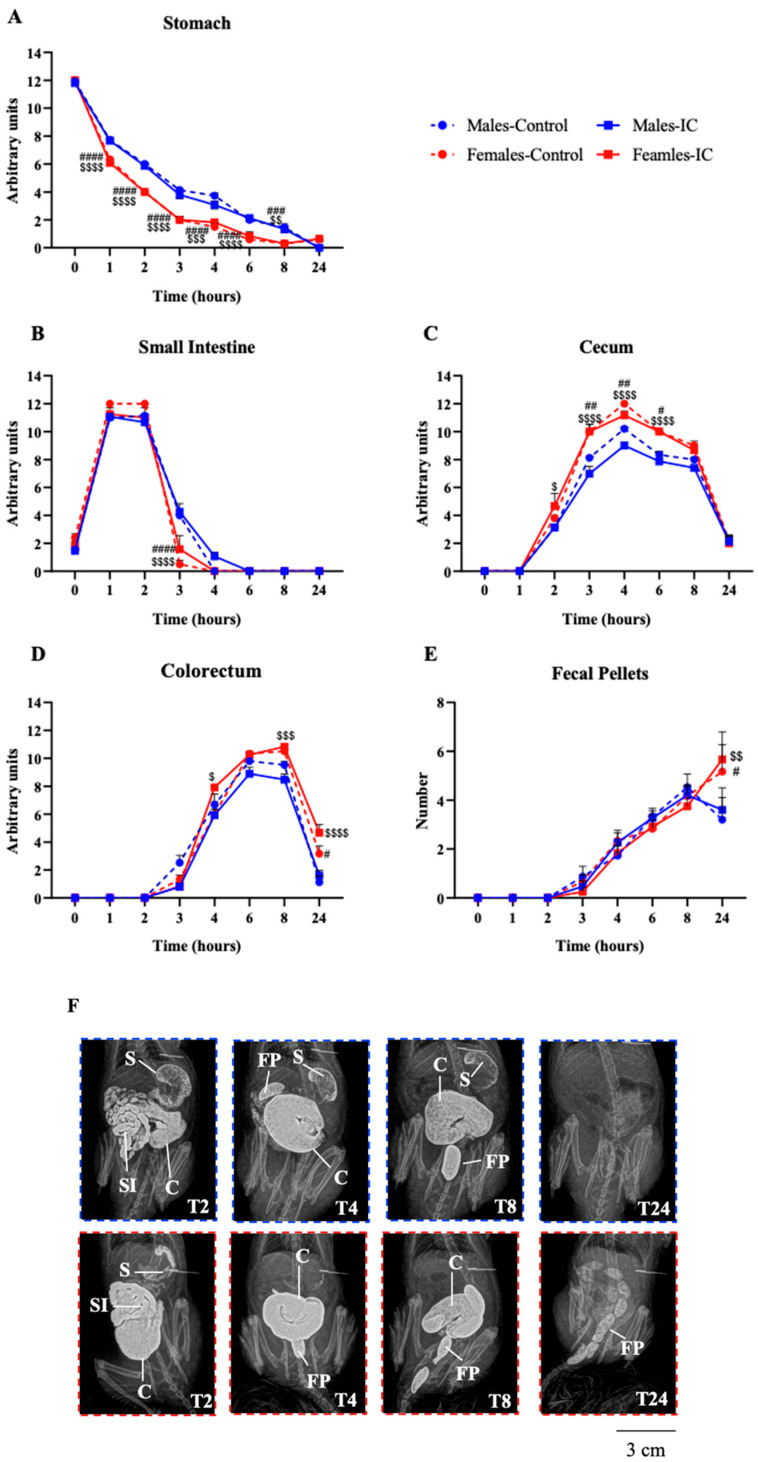
The effect of instant cascara (IC) beverage on gastrointestinal transit of male and female rats was evaluated radiographically after barium administration. Gastrointestinal motility was evaluated in four experimental groups, distributed according to sex and the administered beverage (IC or water): Males-Control, Males-IC, Females-Control, and Females-IC. The values obtained at each time point in the semiquantitative analysis for the stomach (**A**), small intestine (**B**), cecum (**C**), and colorectum (**D**) are depicted, as well as the number of fecal pellets within the colorectum (**E**). Representative X-rays obtained at T2, T4, T8, and T24 from control males (blue frame, **upper panel**) and control females (red frame, **lower panel**) are shown in (**F**). Data represent mean ± SEM (standard error of the mean). *n* = 12 animals per group. Sex-dependent statistically significant changes: # *p* < 0.05, ## *p* < 0.01, ### *p* < 0.001, #### *p* < 0.0001 (Females-Control vs. Males-Control); $ *p* < 0.05, $$ *p* < 0.01 $$$ *p* < 0.001, $$$$ *p* < 0.0001 (Females-IC vs. Males-IC). Two-way ANOVA followed by Bonferroni’s post-hoc test.

**Figure 4 nutrients-16-00065-f004:**
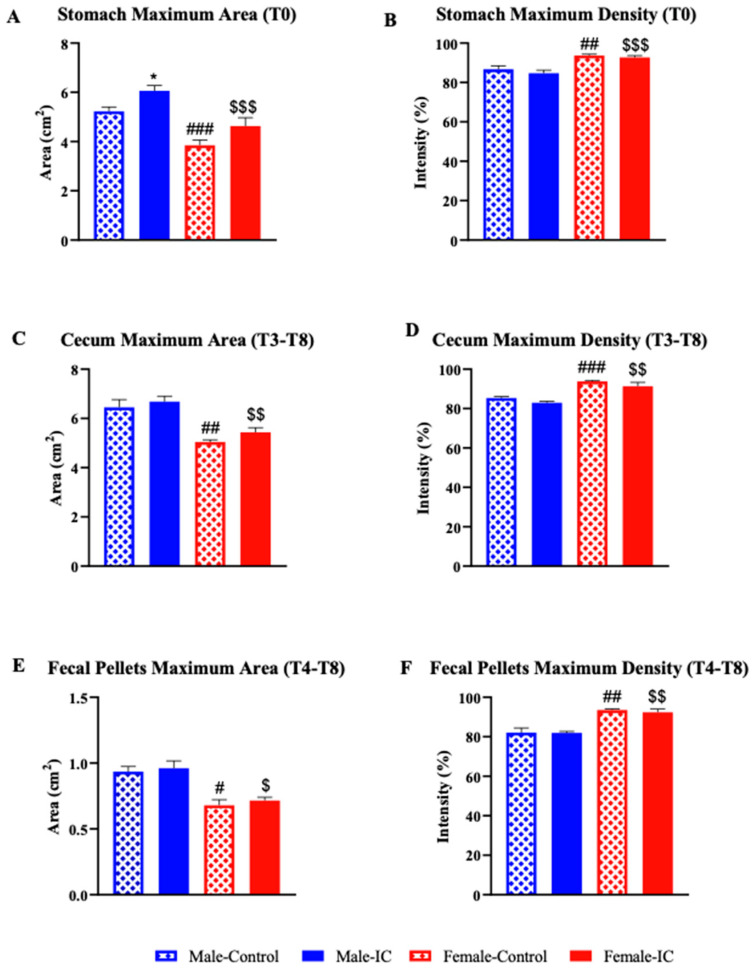
The effect of instant cascara (IC) beverage on the morphometric and densitometric analysis of gastrointestinal organs 3 weeks after IC administration. Area (**A**,**C**,**E**) and density (**B**,**D**,**F**) of the stomach (**A**,**B**), cecum (**C**,**D**), and fecal pellets (**E**,**F**) were measured with ImageJ from the X-rays obtained at 0, 1, 2, 3, 4, 6, 8, and 24 h after barium administration in four experimental groups, distributed according to sex and the administered beverage (IC or water): Males-control, Males-IC, Females-control, and Females-IC. The maximum values for area and barium density were obtained at T0 for the stomach and after averaging the values obtained at T3–T8 for the cecum and at T4–T8 for the fecal pellets. The study was performed 3 weeks after IC or water exposure. Data represent mean ± SEM (standard error of the mean). *n* = 12 animals per group. Sex-dependent statistically significant changes: # *p* < 0.05, ## *p* < 0.01, ### *p* < 0.001, [Females-control vs. Males-control]; $ *p* < 0.05, $$ *p* < 0.01, $$$ *p* < 0.001 (Females-IC vs. Males-IC). Beverage-dependent statistically significant changes: * *p* < 0.05 (Males-IC vs. Males-Control). One-way ANOVA followed by Bonferroni’s post-hoc test.

**Figure 5 nutrients-16-00065-f005:**
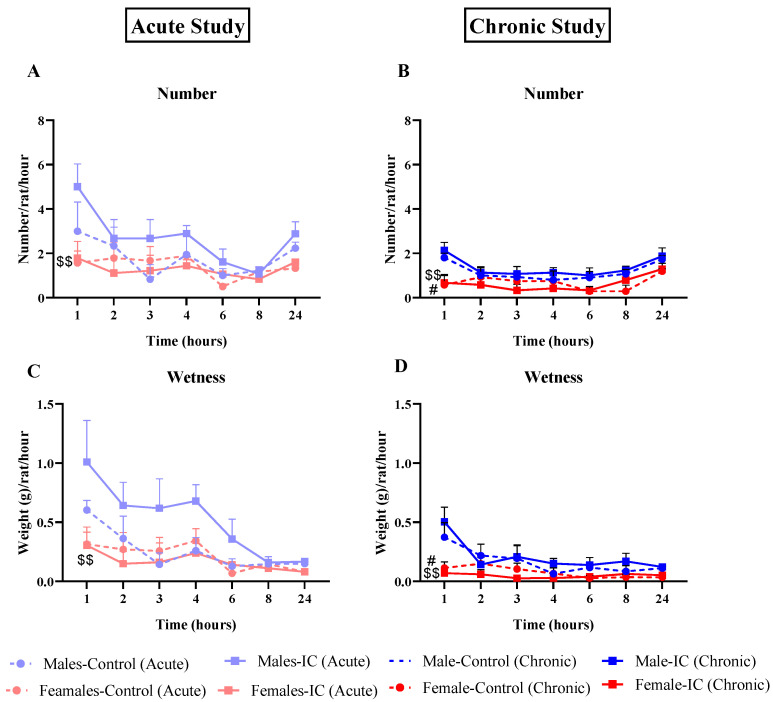
Effect of instant cascara (IC) beverage on the number and wetness of feces of male and female rats. The number of fecal pellets (**A**,**B**) and the differences between wet and dry weight (Wetness, (**C**,**D**)) are represented for the acute (**A**,**C**) and chronic (**B**,**D**) studies. There were four experimental groups, distributed according to sex and the beverage administered (IC or water): Males-Control, Males-IC, Females-Control, and Females-IC. Data represent mean ± SEM (standard error of the mean). *n* = 9 (**A**,**C**) or 12 (**B**,**D**) animals per group. Sex-dependent statistically significant changes: # *p* < 0.05 (Females-Control vs. Males-Control); $$ *p* < 0.01 (Females-IC vs. Males-IC). Two-way ANOVA followed by Bonferroni’s post-hoc test.

**Figure 6 nutrients-16-00065-f006:**
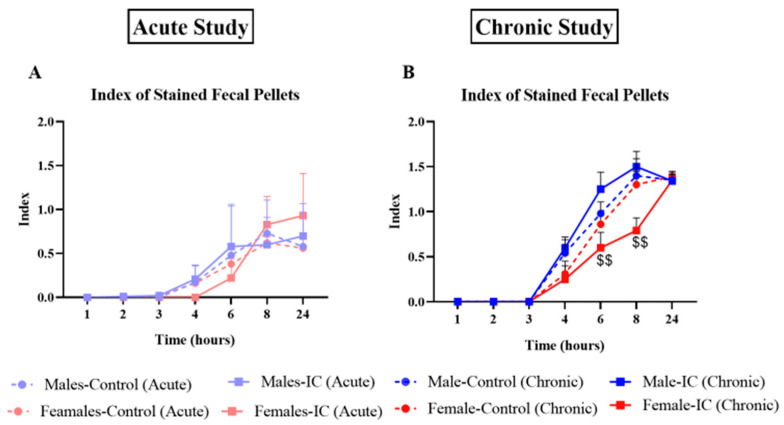
Effect of instant cascara (IC) beverage on the index of stained feces of male and female rats after 1 day ((**A**), acute study) or 3 weeks ((**B**), chronic study) of IC exposure. There were four experimental groups, distributed according to sex and the beverage administered (IC or water): Males-Control, Males-IC, Females-Control, and Females-IC. Data represent mean ± SEM (standard error of the mean). *n* = 9 (cohort 2) or 12 (cohort 1) animals per group. Sex-dependent statistically significant changes: $$ *p* < 0.01 (Females-IC vs. Males-IC). Two-way ANOVA followed by Bonferroni’s post-hoc test.

**Table 1 nutrients-16-00065-t001:** Effect of instant cascara on the anhedonic behavior of male and female rats measured by means of a splash test.

	Males-Control	Males-IC	Females-Control	Females-IC
Latency (s)	111 ± 13.8	98.4 ± 12.5	71.1 ± 5.6	93.8 ± 17.7
Duration (s)	37.5 ± 6.4	42.4 ± 6.0	45.3 ± 8.8	51.3 ± 11.4

Four experimental groups, distributed according to sex and the administered beverage (IC or water), were used: Males-Control, Males-IC, Females-Control, and Females-IC. Data represent mean ± SEM (standard error of the mean) of *n* = 12 animals per group. One-way ANOVA.

**Table 2 nutrients-16-00065-t002:** Effect of instant cascara (IC) on the exploratory behavior of male and female rats.

		Males-Control	Males-IC	Females-Control	Females-IC
Number of crossed squares	Total	209.6 ± 17.4	228.3 ± 15.3	251.42 ± 17.9	285.3 ± 22.3
	Internal (%)	3.38 ± 0.6	3.7 ± 0.6	4.6 ± 0.7	4.2 ± 0.6
	External (%)	97.7 ± 1.4	96.4 ± 0.6	95.4 ± 0.7	95.8 ± 0.6
Time (s)	Holes	27.1 ± 5.3	35.1 ± 5.9	45.4 ± 5.7	38.83 ± 5.3
	Grooming	31.3 ± 6.5	23.40 ± 6.2	33.5 ± 7.2	30.8 ± 7.7
	Standing on hind-limbs	76.3 ± 12.6	76.1 ± 8.7	82.3 ± 9.8	122.4 ± 13.8 ($)
Number of feces		1.20 ± 0.37	1.13 ± 0.39	0.42 ± 0.35	0.58 ± 0.3

Four experimental groups, distributed according to sex and the administered beverage (IC or water), were used: Males-Control, Males-IC, Females-Control, and Females-IC. Data represent mean ± SEM (standard error of the mean). *n* = 12 animals per group. Sex-dependent statistically significant changes: $ *p* < 0.05 (Females-IC vs. Males-IC). One-way ANOVA followed by Bonferroni’s post-hoc test.

**Table 3 nutrients-16-00065-t003:** Effect of instant cascara (IC) beverage on the anxiety behavior of male and female rats.

	Males-Control	Males-IC	Females-Control	Females-IC
Closed arms (s)	75.8 ± 17.4	55.7 ± 15.1	56.2 ± 16.8	46.2 ± 8.3
Open arms (s)	166 ± 18.6	175 ± 23.0	195 ± 18.6	189 ± 7.2
Entries to open arms (number)	6.11 ± 0.9	5.56 ± 0.9	6.3 ± 1.1	5.3 ± 0.5

Four experimental groups, distributed according to sex and the administered beverage (IC or water), were used: Males-Control, Males-IC, Females-Control, and Females-IC. Data represent mean ± SEM (standard error of the mean). *n* = 9 animals per group. One-way ANOVA.

## Data Availability

The data presented in this study are available on request from the corresponding author and will also be available at the corresponding authors’ institutional repository (URJC).

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
