# Peer review of "Evaluation of the Effects of Instant Cascara Beverage on the Brain-Gut Axis of Healthy Male and Female Rats"

_nutrients, 2023, doi:10.3390/nu16010065_

Round 1

Reviewer 1 Report

Comments and Suggestions for Authors

In the manuscript submitted to me for review entitled "Evaluation of the effects of Instant Cascara beverage on the brain-gut axis of healthy male and female rats the authors Paula Gallego-Barceló, Ana Bagues, David Benítez-Álvarez, Yolanda López-Tofiño, Carlos Gálvez-Robleño, Laura López-Gómez, María Dolores Del Castillo and Raquel Abalo investigated the effect of INSTANT CASCARA (IC, a sustainable beverage obtained from dried coffee cherry pulp) on general health and parameters of the brain-gut axis in healthy female and male rats.

The study brings important information related to the use of a drink made from dried coffee cherry pulp. Such research is extremely necessary when introducing a new food product or beverage into the commercial network. It should be known whether the given product causes any side effects in the body.

In the study, the required ethical norms for conducting experiments with laboratory animals were observed. The methods used are well described. The obtained results are presented using 6 well-designed and structured figures and 3 tables. The described results fully correspond with the conclusions made by the authors.

To support their research, the authors used 71 references that present information from studies spanning the past five decades. Almost 2/3 of the total number of references are from the last 5 years, which shows that other teams are also working on similar topics and the issue is relevant. I did not notice any redundant self-citations, all references used are relevant and necessary for the preparation of the manuscript.

My remarks and recommendations to the authors are:

1. I have questions related to the methodological part regarding the concentration of the product used. 10 mg/ml of test product was reported to be used.

a) how this dose was determined - in some initial studies or according to literature data - is not clear from the description of the methodology;

b) is this the final concentration contained in the water or from this concentration is it put into the drinking water?

2. Female individuals have been reported to increase fluid intake as a result of drinking. What is meant? In the animals a) there is additional water not containing the test product and the intake of this water is increased or b) the only water available in the cells of the test animals contains IC and this leads to a greater intake of the water containing the test product? If it's the second option, wouldn't that suggest that female subjects might like the taste of IC more, or that its use leads to some form of "addiction" to the taste of IC?

3. I think it would be better if figure 3F was given as a separate figure with subfigures and each subfigure described what it represents.

4. In the References section, some references do not list all the authors (for #7, 17, 37, 53, 61, 62 and 66). I think it would be better for readers to add all the authors.

5. In reference #16, the year of publication is not indicated. Let the authors do a check and add the year.

Reviewer 2 Report

Comments and Suggestions for Authors

The authors provide an excellent study on the effects of instant cascara in experimental animals. The results are well documented and the conclusions are based in the data.

The following revisions might be considered:

- line 28: what is the dose in relation to body weight? Is this comparable dose to humans drinking IC?

- consider deleting abstract lines 31-36 as the expected sex differences are not so interesting. The abstract is very long and should be focused on the most important results.

- line 43: repeats studied dose from line 28

- abstract and conclusion: is there are NOAEL that could be derived?

- Line 67: I do not think that IC would be possible for the claim "high in fiber" (the fiber is mostly discarded by the aqueous extraction?)

- Line 92: fullstop missing

- Line 121: can the composition be shortly detailed here?

- Line 134: can a short rationale for the concentrations be provided? Is this comparable to human dose?

- Figure 6: the resolution is very low compared to the other figures

- all figures: consider deleting the title text at the top. This should be in the legend.

- general comment: did you measure liver enzymes or any other biochemical parameters?
